# One Health Ethics and the Ethics of Zoonoses: A Silent Call for Global Action

**DOI:** 10.3390/vetsci11090394

**Published:** 2024-08-27

**Authors:** Jeyver Rodriguez

**Affiliations:** 1Department of Applied Ethics, Temuco Catholic University, Temuco 4780000, Chile; jeyver.rodriguez@uct.cl; 2Cape Horn International Center for Global Change Studies and Biocultural Conservation (CHIC), Cabo de Hornos 635000, Chile

**Keywords:** one-health, zoonotic diseases, one bioethics, animal slaughter, zoonoethics, AMR, complexity, bioethics, public policy

## Abstract

**Simple Summary:**

This paper reviews key issues regarding the spread of diseases that affect both humans and animals, known as zoonotic diseases. These diseases, which make up about 70% of all new and existing diseases, are increasingly interconnected with factors like the environment, society, and economics. The term “zoonoses” is discussed beyond just veterinary medicine, emphasizing its complex nature influenced by human activities and climate change. Bioethical principles and strategies for preventing these diseases are proposed. A case study on animal slaughter during disasters highlights ethical challenges in how we manage health across species, leading to discussions on “zoonoethics”. The paper also explores how these diseases emerge and the ethical and political issues around antimicrobial resistance (AMR), with recommendations for tackling AMR. The management of these diseases will require the adoption and acceleration of the “One Health” approach, which recognizes the interconnectedness and interdependence between human, animal, and environmental health.

**Abstract:**

This paper presents a critical review of key issues related to the emergence of new networks for the spread of zoonotic diseases amid the mass extinction of species. Zoonotic and infectious diseases account for approximately 70% of new and existing diseases affecting humans and animals. The initial section argues that the term “zoonoses” should not be confined to single-cause events within veterinary medicine. Instead, zoonoses should be viewed as complex, systemic phenomena shaped by interrelated factors, including environmental, sociocultural, and economic elements, influenced by anthropogenic climate change. The second section presents bioethical principles and potential strategies for those engaged in zoonotic disease prevention. The third section uses the slaughter of animals in disaster settings as a case study to illustrate the need for further clarification of normative and interspecies justice conflicts in One Health ethics. This section concludes with an outlook on “zoonoethics”. Section four develops the analysis of the interlinked elements that trigger zoonoses and examines antimicrobial resistance (AMR) from an ethical and political standpoint, concluding with policy recommendations for addressing AMR. Section five offers a critical reflection, integrating contributions from zoonoethics, human ecology, and the ecotheological turn. Finally, section six concludes with a call to action and policy recommendations for an inclusive, intercultural, and gender-sensitive One Health approach.

## 1. Introduction: Incremental Risks of Zoonotic Diseases

Zoonotic pathogens can spread to humans through contact with domestic, farm, or wild vertebrate animals [1]. However, invertebrates and other intermediate hosts can intervene in the complex network of emergence and transmission of zoonotic diseases [2,3,4,5]. Some zoonotic diseases of animal origin are severe acute respiratory syndrome (SARS), Ebola and, recently, SARS-CoV-2. As shown by Jones et al. [6] (p. 990), a large proportion of emerging and re-emerging infectious diseases are zoonotic in origin (ERIDs) are zoonoses (60.3% of EIDs), and more than 70% originate from wild animal trafficking [7,8,9,10,11,12,13], increasingly close contact with farm and wild animals, intensive agriculture, unsustainable global food systems, and crimes against biodiversity [14,15,16,17].

The objective of this article is twofold. On the one hand, I want to join the recent call to action to accelerate the operationalization and implementation of the One Health approach (OH), made by the Quadripartite organizations working on One Health [18]. On the other hand, the article emphasizes that the tools and methods of multispecies justice (MSJ) and ethics of zoonoses can help resolve conflicts between competing moral claims that arise in relation to both human and non-human life. This approach can help address the challenges facing the OH framework. In this article, I address the more established definition of One Health offered by the One Health High-Level Expert Panel (OHHLEP): “One Health is an integrated, unifying approach that aims to sustainably balance and optimize the health of people, animals, and ecosystems. It recognizes the health of humans, domestic and wild animals, plants, and the wider environment (including ecosystems) are closely linked and interdependent. The approach mobilizes multiple sectors, disciplines, and communities at varying levels of society to work together to foster well-being and tackle threats to health and ecosystems, while addressing the collective need for healthy food, water, energy, and air, taking action on climate change and contributing to sustainable development” [18] (p. 2).

Today, there is ample evidence that working transdisciplinary and collaboratively across sectors to accelerate and implement the One Health approach will help build long-term capacities for ‘preparedness, resilience, mitigation, and effective prevention of future epidemics including zoonotic and (re-)emerging infectious diseases and non-communicable diseases linked to environmental risk factors, antimicrobial resistance (AMR) and climate adaptation’ [19,20,21] (p. 3). In this article, I suggest that the emergence and re-emergence of epidemic outbreaks is strongly related to the drivers of anthropogenic climate change, such as abrupt changes in land use, deforestation, ecological racism, ecocide, environmental crimes, and the accelerated loss of biodiversity. Although there is still a strong debate between the scope, relevance, and epistemic and normative potential of the One Health and Planetary Health approaches, I emphasize that the best way forward is to articulate these approaches and bring them into dialogue to work together [22,23,24].

According to the above, in this work, I would like to highlight two primary objectives for accelerating and implementing the OH approach. The first priority can be stated as follows: It is imperative that governments, institutions, and civil society take immediate action to halt the systematic destruction of wildlife and put an end to the ongoing ecocide on the planet. Only through precautionary and holistic measures can we hope to see a reduction in the incidence of emerging and re-emerging zoonotic diseases [25,26,27,28,29,30]. As evidenced by a substantial body of scholarly research, there is a robust correlation between the loss of animal species, environmental crime, ethnocide, and the mechanisms involved in the emergence and spread of zoonotic diseases. Secondly, in order to mitigate the risks associated with emerging infectious diseases (EIDs) and zoonoses in the context of mass species extinction, it is imperative to improve inter-epistemic and transdisciplinary communication in order to develop a holistic and non-anthropocentric perspective of EIDs and zoonoses [31,32,33,34].

Zoonoses and emerging infectious diseases should not be understood as single-cause events reduced to the world of veterinary medicine but as systemic ecosocial phenomena of increasing complexity that are produced by the confluence of environmental, sociocultural, and economic factors linked to the anthropogenic climate change [23,35,36,37,38,39], and the complex nexus between virus flows, global capitalism, and ecological racism [40,41,42,43,44,45]. It is evident that infectious diseases cannot be considered as a mere “biological phenomenon”; rather, they are intricately intertwined with the dynamics of modern technoscientific societies and the systematic annihilation of biocultural diversity [40,46,47,48,49]. To understand the new infectious diseases, it is necessary to critically examine the dynamics and flows of the global economy, which creates a socio-political climate of structural pathogenesis [50,51,52,53]. Figure 1 illustrates the network of factors associated with the emergence and spread of zoonoses.

Zoonotic diseases can emerge and spread rapidly among populations, overwhelming the capacity of institutions responsible for epidemiological surveillance, due to globalized flows of biological agents—human, animal, and plant. Accelerated population growth and increased international travel and global trade also play a key role in the emergence of zoonotic risks [3,22,54,55,56,57,58,59,60]. Other anthropogenic drivers that increase the risks of zoonoses are pollution of the seas, overfishing, excessive use of antibiotics in the global food production chain, and the advance of “green extractivism” [61,62]. Recent studies even show a strong link between plastic pollution and novel infectious diseases [63]. This link is particularly clear in the case of arthropod-borne diseases, which raise a variety of ethical concerns (e.g., dengue, chikungunya, Zika, Japanese encephalitis, leishmaniasis, and Chagas disease) [64,65].

Industrial poultry, swine, and livestock farms use excessive doses of antimicrobials at various stages of animal production. This practice puts them at risk of generating antimicrobial resistance chains and becoming hotspots for the spread of emerging and re-emerging zoonotic diseases. In fact, the call to “Curb the silent pandemic of AMR” is one of the main red traffic lights that we find in the One Health Joint Plan of Action, 2022–2026 [66]. In fact, the ethical and prudential use of bactericidal and antimicrobial preparations has ceased to be a question for experts and is becoming one of the great ethical imperatives for the construction of viable and sustainable health systems [66,67,68,69,70,71,72,73,74,75,76].

The widespread and prolonged use of antimicrobials in the different stages of intensive animal agriculture represents one of the main drivers for the development of resistant bacterial populations. The ethical use of antimicrobials has strong implications for both human ethics and animal ethics, although this difference is becoming increasingly blurred, and it is noted that the prudent management of zoonoses requires transdisciplinary approaches and interdisciplinary discussions, including socio-political and bioethical issues [77,78].

Another factor that increases the risk of zoonotic diseases is the unsustainable transport of livestock and poultry, since these farm animals are often transported in unsanitary conditions over long distances, which deepens both animal abuse and the risks of spreading zoonotic diseases [79,80,81]. In other words, the crisis of the global food system, which has led to a decline in the intake of healthy foods with high nutritional value, overlaps with other crises, such as the loss of biodiversity, the water crisis, floods, lack of water treatment plants, lack of hygiene and sanitation, and the contextualized biodiversity crisis [82,83,84,85,86,87,88,89].

As Vercauteren et al. [90] (p. 3) show: “The livestock compartment and its interfaces with humans and wildlife appeared after domestication. These epidemiological interfaces have constituted opportunities for horizontal transmission between species and a new space for evolution, emergence, and maintenance of pathogens”. Due to anthropogenic stressors and accelerated changes driven by intensive agriculture, new chains of pathogen transmission have emerged that can range from human host epidemics with zoonotic genetic source contribution to animal host epidemics with human genetic source contribution. These chains of pathogen spread can involve primary reservoirs (wildlife), domestic animals as potential reservoirs, intermediate hosts, vector-borne, foodborne, and human hosts [91,92,93,94,95,96]. Unpredictable changes in the dynamics of the spread of endemic zoonotic diseases raise major ethical concerns about the re-emergence of new viral variants. A case in point is the recent WHO declaration of pox as a public health emergency of international concern. This disease, caused by an orthopoxvirus, was first detected in humans in the Democratic Republic of the Congo in 1970 and has re-emerged as variant 1b since 2022. [97].

The close relationships that humans develop with both companion and wild animals are evident in various instances. Beekeepers form deep connections with insects, while individuals in indigenous communities, as well as divers, filmmakers, and researchers, cultivate friendships and empathy with mammals like whales and sharks, as well as cephalopods, primates, and elephants. Several factors are involved in the spread of pathogens, including wildlife, human, and domestic animals, food systems, vectors, bacteria, parasites, fungi, and viruses, among others. While there are situations where nonhuman animals and humans coexist in relative harmony, the increased activity in biotechnology, science, and capitalist economy coupled with the rapid loss of wildlife and biodiversity, has led to close relationships between nonhuman animals and humans becoming a contributing factor to increased vulnerability and zoonotic risks. Zoonotic diseases such as avian influenza, rabies, Ebola, Rift Valley fever, Nile virus, food-borne diseases, and antimicrobial resistance pose a threat to the future survival and sustainability of human and non-human communities [20,66].

The type of care, relationships, and inter-connections established with companion, farm, and wild animals result in incremental dependency and susceptibility to zoonotic diseases. The entangled communities of human, animal, plant, and microbial agents is a concept that can be supported by critical animal studies, ecofeminism, and symbiotic biology [98,99,100,101,102]. From the perspective of bodily ecology, it is evident that our practices of care and conviviality are deeply embedded in dynamic and complex exchanges with other species, as well as with the genes, memories, practices, and knowledge of our ancestors. Therefore, initiatives to prevent and mitigate infectious diseases must be contextualized and linked to the knowledge and culturally rooted practices of entangled communities.

As I will show in this article, pathogen transmission networks are diverse and complex, involving interconnected relationships and complex interdependencies between human and nonhuman animals and the environment. Influenza and coronaviruses are two typical cases of reverse zoonoses (zooanthroponoses) [103,104,105]. The chains of spread of zoonotic diseases can be direct: by being in contact with saliva, blood, urine, or mucous membranes of an infected domestic or wild animal, or indirect, by coming into contact with areas, surfaces, or objects that have been contaminated by germs [106]. The increasing number of people who choose to live with and keep wild animals as pets has made it more complex and increased the risk of producing zoonoses in various ways [107,108,109].

Wilcox & Steele [110] argue that epidemic diseases manifest in the form of sporadic outbreaks that are inherently unpredictable in terms of their spatial, temporal, and severity characteristics. In contrast, emerging zoonoses represent a distinct category of zoonotic diseases that have recently emerged within a given population or have experienced a marked increase in incidence or geographical distribution. Löscher et al. [4] shows that the factors that explain changes in the dynamics of the spread of emerging and re-emerging pathogens are complex and range from macro-level factors such as population growth and climate change to micro-level factors linked to diet, hygiene, sanitation, and personal disease risks. Major emerging and re-emerging infectious agents in the past decade include Ebola virus in Africa, Middle East respiratory syndrome coronavirus in the Middle East, and Zika, chikungunya, yellow fever, and dengue viruses in the Americas. Another hot topic in One Health is neglected tropical diseases (NTDs). According to WHO [111], NTDs “are a diverse group of conditions caused by a variety of pathogens (including viruses, bacteria, parasites, fungi and toxins) and associated with devastating health, social and economic consequences”. These diseases have traditionally been associated with poor countries and regions with wide health disparities. However, this dynamic is changing. As emphasized by the European Centre for Disease Prevention and Control (ECDC): “Europe is already seeing how climate change is creating more favourable conditions for invasive mosquitos to spread into previously unaffected areas and infect more people with diseases such as dengue” [112]. Climate change affects vectors differently; for example, mosquitoes and ticks’ geographical expansion depends on temperature, wind, humidity, and rainfall [113]. Reports indicate increased malaria prevalence due to climate change in Zimbabwe, Brazil, and Colombia, among others countries [114,115,116].

In Europe, the main concern is mosquito-borne viruses. These include West Nile, dengue and chikungunya [117]. Fortunately, these countries have powerful tools, robust health systems, and the resources to develop AI tools such as The Surveillance Atlas of Infectious Diseases, which can analyze large amounts of data on the risks of mosquito-borne viruses and provide early warnings. As the dynamics related to abrupt changes in land use, tourism, biodiversity loss, and wildlife trafficking deepen, vector-borne diseases increase due to the confluence of environmental, socio-cultural, and abiotic changes in temperature, air, and humidity, among other factors. The phenomenon of global warming, a consequence of climate change, provides an environment conducive to the proliferation of disease vectors that transmit and spread malaria and dengue.

Nevertheless, vector-borne zoonotic disease outbreaks may occur in regions and countries where communities and governments lack the necessary resources and capacity to conduct comprehensive epidemiological surveillance, thereby hindering the ability to trace and control outbreaks in a timely manner.

Figure 2 shows the interrelationship of events that trigger the onset of infectious diseases. There is strong evidence of an interdependence between the global impacts of climate change and increases in local and regional phenomena such as flooding, increased humidity and temperature. As shown by Gibb et al. [118], zoonotic host diversity increases as ecological relationships are disrupted by a wide range of economic and human activities. They “show that land use has global and systematic effects on local zoonotic host communities”. As Mahon et al. [119] (p. 234) states, “The fact that many global change drivers increase zoonotic parasites in non-human animals and increase all parasites in wild animals suggests that anthropogenic change might increase the occurrence of parasite spillover from animals to humans and, therefore, also pandemic risk”.

To understand the complexity of the interdependencies and linkages behind the long chains of zoonotic disease spread, we must continue to deepen the holistic perspective of One Health, which seeks to build bridges to address all risks and vulnerabilities that arise in overlapping communities of life [120,121]. In the context of ongoing technological change, it is urgent to rethink the connections between ethics, climate change, technologies, and agricultural practices [122]. The relationships between endemic and re-emerging zoonotic diseases and the societal and anthropogenic drivers of climate change are becoming increasingly complex [123,124].

Moreover, international transportation and global trade of vast quantities of food, animals, and people are key factors in pathogen transmission networks. Viruses, microorganisms, and pathogens spread as quickly as people do in a hyper-connected and interdependent world. As ref. [125] (p. 195) states: “In modern times, air travel resulted in the importation of severe acute respiratory syndrome (SARS) coronavirus to 27 countries before transmission was halted”. Indeed, advances in health systems, sanitation, and tools for disease detection and tracking have led to reductions in overall mortality and morbidity from infectious diseases, particularly in high-income countries. As usual, however, scientific progress has not been evenly distributed around the world. As Baker et al., [125] (p. 193), show: “infectious disease burden remains substantial in countries with low and lower-middle incomes, while mortality and morbidity associated with neglected tropical diseases, HIV infection, tuberculosis and malaria remain high. […] This points to a possible new era of infectious disease, defined by outbreaks of emerging, re-emerging, and endemic pathogens that spread quickly, aided by global connectivity and shifted ranges owing to climate change”.

The nonlinear and increasingly complex events that trigger infectious diseases in the Age of Extinction are such that they cannot be understood from a single discipline or with narrow methods. While there is evidence that poverty, poor sanitation, and inadequate sewage and water systems provide a breeding ground for the emergence and spread of infectious diseases such as malaria and cholera, other diseases like those caused by SARS-CoV-2 may be related to increased affluence, urbanism, population growth, and wildlife trade chains in megacities. The socioeconomic factors affecting the spatial distribution of potential risk zones must continue to be carefully studied [126,127,128,129,130,131].

## 2. Zoonoethics, Global Bioethics, and One Health: We Need “One Bioethics”?

In this section, I introduce a set ethical principles and potential strategies that zoonoethicists concerned with preventing emerging zoonotic diseases should consider. These guidelines aim to inform public policy and guide decision-making for communities, citizens, decision-makers, and stakeholders. On the one hand, zoonoethics, understood as a field of research in applied ethics and veterinary medicine, can promote rational, evidence-based discussions to exercise prudent judgment and public deliberation based on constructive conflict to resolve normative disputes about “conflicting value claims” linked to the health of humans, animals and environments. On the other hand, zoonoethics must be rooted in the precautionary principle and provide normative and descriptive knowledge to inform decision-makers and interested parties in the construction of public policies aimed at preventing ERIDs. While not a definitive solution, the precautionary principle can provide a framework for navigating disputes over competing values and assets and for developing proactive measures to anticipate and assess the risks associated with ERIDs in uncertain contexts [132,133,134,135,136,137] The precautionary principle, when considered in conjunction with epistemic deliberation and prudent judgment, can inform the theory of change that guides epidemiological judgments and decisions at the One Health initiative [29,30,138,139,140,141].

In light of the statements above, it is possible to sustain a third claim: Zoonoethics can serve as a “bridge” to unite diverse epistemic fields and establish transdisciplinary and inter-paradigmatic debates between scientific, religious, cultural, and ecosocial pluriverses. In the context of One Health initiatives targeting infectious diseases, it is imperative to ensure cultural sensitivity and to avoid underestimating the influence of racial, ethnic, religious, and cultural diversity on public health strategies aimed at curbing the transmission of zoonotic diseases [142,143,144,145,146].

Zoonoethics, at the epistemic level, serves as a critical framework for integrating diverse knowledge systems and disciplines. By fostering dialogue between fields such as veterinary medicine, biomedical sciences, animal agriculture, epidemiology, antimicrobial resistance, and global bioethics, zoonoethics facilitates a holistic understanding of the complex interconnections between human, animal, and environmental health. This interdisciplinary approach also incorporates insights from human ecology, conservation sciences, biocultural ethics, and environmental humanities, promoting a more holistic and critical perspective. Such integration is essential for addressing the multifaceted challenges posed by zoonotic diseases and ensuring that ethical considerations are deeply embedded in the development of effective and equitable health strategies [147,148,149,150]. 

In this article, I introduce the term “zoonoethics” to highlight the necessity for an interdisciplinary field of work that integrates resources derived from normative ethics and veterinary medicine. The term “zoonoethics” can be understood as an interdisciplinary field that addresses ethical concerns related to the intersection of human, animal, and environmental health. The term “zoonoethics” follows a similar path to “global bioethics”, which strives to integrate insights from biological and human sciences for future sustainability [151]. Zoonoethics can be understood as a subset of One Health ethics, encompassing a broader range of concerns for human, animal, and ecosystem health, including concerns for future generations. These concerns include the ethics of antimicrobial resistance, ethics of species conservation, nonhuman ethics, wildlife ethics, and the ethics of global food. Therefore, the zoonoethics I am attempting to delineate can be regarded as a subfield of One Health ethics, rather than as another “silo” [152]. Potter’s vision facilitated the integration of public health concerns with broader issues pertaining to the sociocultural determinants of health and global challenges such as pollution, biodiversity loss, technological change, and the impact of accelerated population growth on future sustainability [153,154,155,156,157]. By situating zoonoethics within the One Health approach, it takes a more holistic view of the ethical challenges arising from the interaction between humans, animals, and environments.

The zoonoethics should be focused on the management and prevention of zoonotic diseases in three specific areas:(i)Primary prevention: zoonoethics, in close collaboration with the One Health ethics [29,30,139,158,159,160,161], can offer descriptive and epistemic resources for transforming our understanding *of* and interconnections with nonhuman beings and environments (both natural and built), as well as for clarifying our relationships with them. Primary prevention aims to anticipate risks before they become fully manifest. As Plowright et al. [162] (p. 2) show: “Primary pandemic prevention is the set of actions taken to reduce the risk of pathogen spillover from animals to humans, focusing on processes upstream of the spillover event”. The first preventive measure to reduce zoonotic risks and the spread of viruses with pandemic potential is to practice ecological wisdom related to the holistic protection of species and the care of the biodiversity of ecosystem hotspots. As highlighted by Vora et al. [163] (p. 420), “tropical and subtropical forests must be protected”.(ii)Secondary prevention: Primary prevention is, by definition, anticipatory, while secondary prevention focuses on the implementation of specific measures, such as early detection, vaccines, improved health systems, and drug therapy, but these are established in the ongoing process of preventing the outbreak from becoming an epidemic or pandemic [164,165]. At this stage, zoonoethics can provide a framework for setting standards and guiding policymakers on what actions are most effective, fair, and ethically relevant to prevent the escalation of the outbreak. One of the paradoxes of primary prevention is that it is largely undervalued as an effective strategy for responding to pandemic risks, while the great paradox of secondary prevention is that many interventions focus on implementing public health measures that may have adverse effects in order to contain the spread of spillover. However, there is sufficient evidence to support the idea that the most cost-effective, wise, and politically relevant strategy for preventing future pandemics is to invest in prevention and capacity building in the context of future sustainability [2,20,166,167].(iii)Antimicrobial Stewardship (AMS): Zoonoethics can help to address the problems associated with antimicrobial resistance and to develop a more robust and comprehensive approach to AMS [168,169,170]. As Shallcross et al. [76] (p. 4) stresses: “If AMR is allowed to continue unchecked, we may enter a ‘post-antibiotic era’ of medicine, in which treatments from minor surgery to major transplants could become impossible, mortality will rise, and healthcare costs will spiral as we resort to newer, more expensive antibiotics and sustain a greater number of longer hospital admissions”. AMS is an ethical approach that takes very seriously the growing threat of entering a “post-antibiotic era”. Curbing the emergence and spread of these resistant organisms and agents is a high priority not only in epidemiological surveillance and drug development programs but also in inclusive public policies aimed at reducing inequities in access to safe and affordable medicines for all. As Dyar et al. [171] (p. 793) states: “Although antimicrobial stewardship originated within human healthcare, it is increasingly applied in broader contexts including animal health and One Health”. Experts in zoonoethics can significantly contribute to the efforts of antimicrobial stewardship (AMS) in addressing this issue. Zoonoethics can provide valuable insights and frameworks that can enhance the AMS strategies across different domains—animal health, human health, agriculture, and livestock. Firstly, zoonoethics emphasizes the interconnectedness of ecosystems and the ethical considerations that arise from this interconnectedness. By integrating zoonoethics principles, AMS experts can develop more holistic policies that consider the welfare of all species affected by antimicrobial use. This approach aligns with the One Health perspective, which recognizes the interdependence of human, animal, and environmental health. Moreover, zoonoethics can offer tools such as ethical risk assessments and value-based decision-making frameworks. These tools enable stakeholders to evaluate the implications of antimicrobial use and resistance not only from a scientific standpoint but also through an ethical lens, considering the long-term consequences for all species. For instance, ethical risk assessments can help identify practices that may inadvertently contribute to AMR and propose alternatives that are ethically and ecologically sound. My perspective on the problem of antimicrobial resistance is enriched by the multispecies justice approach, which seeks to integrate other species and wildlife into risk analyses [172,173,174].

Finally, one of the “heat issues” that zoonoethics must address is the fair, careful, and ethical treatment of companion animals and farm and wild animals. In our techno-scientific society, it is generally assumed that the slaughter and killing of farm animals is justified in regard to ensuring food security for a voracious and exponentially growing human population. Nevertheless, killing animals without a “proper purpose” and sufficiently legitimate reasons is considered ethically incorrect by some scholars [175,176,177,178,179,180]. However, in many practices such as intensive agriculture, the culling of animals to mitigate biodiversity loss caused by invasive species, and medical research, among others, animal death is not the exception but the norm. There is still an open debate about whether the issue of death and the conditions in which animals are slaughtered, especially on farms, is properly an animal welfare issue [181,182,183,184,185,186]. On the other hand, the issue of euthanasia or “good death” of animals in zoos would be part of a zooethics rather than the field of zoonoethics, which I am attempting to outline [187].

In this work, I will try to approach the question of the appropriate treatment of animals from an alternative approach—that of ecological and global bioethics [153,188,189,190,191,192]. Now, it is necessary to clarify that my vision of bioethics is opposite to the vision of Fiore [193] (p. 316), who maintains that the term environmental bioethics “seems qualified, derivative: a subgenre of biomedical ethics or environmental ethics rather than the ground for both”. My vision of global bioethics, grounded by Potter’s works [194,195] is nearer to that of authors such as Gardiner [196]; Lee [146]; Macklin [144,197]; ten Have [145], and Valera and Rodriguez [154]. In the aftermath of the COVID-19 pandemic, bioethics must broaden its scope to effectively address the complexity of infectious diseases. From a One Bioethics perspective, it is crucial to integrate global and local approaches, building bridges between diverse cultures and epistemologies to tackle global health challenges with an inclusive, deliberative, and collaborative approach. By rediscovering the normative significance and global appeal of bioethics and its interest in nonhuman ethics, global public health, and climate justice, we can build bridges and pathways between multispecies justice, One Bioethics, and zoonoethics [198].

Currently, there is renewed interest in rethinking and expanding the global agenda of bioethics, as it was conceived at the time by Van Rensselaer Potter [192,199,200]. If we recover the globality and multidimensionality of bioethics, we can build a bridging, comprehensive, and inclusive bioethics that serves to deepen dialog, transdisciplinary communication, and cooperation between countries and regions for an early and prudent response to future outbreaks of ERIDs and pandemics [201,202,203,204,205]. Reinvigorating the global scope of bioethics, in light of recent discoveries about the complex mechanisms behind pathogen emergence and transmission, would significantly enhance the traction and operationalization of One Health and Planetary Health approaches [9,23,36,206,207,208]. 

This tension between One Bioethics and grassroots justice claims, or, in other words, between the globality of bioethics and the and context-dependent character of justice claims, is very interesting and complex, but it is beyond the scope of the present paper. Here, I can only offer a few hints. First, One Bioethics, both in its epistemic roots and in its practical scope, should be both local (context-dependent) and global, because the problems it addresses—related to technological change, transhumanism, biomedicine, AI in health care, climate change, biodiversity loos, ERIDs, etc.—are both local and global, and require global action, but they also require deliberative approaches and situated hermeneutics [209,210]. The need to respond to the complexity and uncertainty of the problems we are dealing with is at the heart of the glocalization of bioethics [211]. Moreover, bioethics must be sufficiently dynamic and retain its deliberative and dialogical character to remain relevant across different intercultural universes [212,213,214,215,216,217,218,219,220]. Once again, one of the most challenging issues in One Bioethics requires the elaboration of “bridges” and inter-epistemic dialogues between different epistemologies, value systems, cultures, religions, disciplines, worldviews, and social practices.

Second, whilst it is true that individuals assign varying values to species, nonhuman animals, ecological entities, and artificial objects (e.g., paintings, forests, violins, plants, rivers, lakes, and landscapes), and that such values are contextual and culturally embedded, this should not diminish the globality of bioethics. My argument, therefore, is for a more humble and dialogical conception of One Bioethics, one that remains open to the uncertainties of a changing world [194,221,222,223]. The globality of One Bioethics is also an invitation to create a window for broad collaborative and interdisciplinary work, and there should be room for diverse symbolic universes, values and norms, and ultimate belief systems. Ethical evaluation and deliberation, in short, must be contextualized, but the global vision of vital problems can also serve as an impetus to tighten the thread of ethical reflection, which, as we know, can sometimes be transboundary, interspecies, transgenerational, and intercultural. Of course, the question of cross-cultural moral evaluation is closely related to the very possibility of engaging in inter-paradigmatic dialogues and raises the very serious question of intercultural, interreligious dialogue and tolerance [202,224,225]. The question is to what extent we are willing to accept as legitimately valid ways of evaluating and assigning different kinds of moral, aesthetic, and religious values to different objects or entities from other cultures, and how willing we are to recognize and value non-Western symbolic universes and value and belief systems.

As Gardiner [196] (p. 571) emphasizes, “conventional bioethics has largely failed to engage, and so is left mainly to contribute to damage limitation, emergency management, and redress”. A way to overcome this limitation and avoid falling into the vices of a superficial and overly technical bioethics that only limits itself to acting when it is too late, but is incapable of providing normative, descriptive knowledge and practical wisdom to guide decision-making and anticipate risks, is to move toward what some authors have called “One Bioethics” [226,227] and One Health ethics [29,30,228]. One Bioethics and One Health ethics could offer valuable normative and epistemic resources to prudently address the risks derived from the emergence and spread of EIDs and zoonoses.

Table 1 presents some bioethical and zoonotic ethics guidelines to face zoonotic diseases. These guidelines aim to inform public policy and guide decision-making for communities, citizens, decision-makers, and stakeholders. The table below outlines practical guidelines for addressing infectious diseases, emphasizing surveillance, interdisciplinary research, public education, and sustainable policies. Effective crisis management, international cooperation, One Health initiatives, ethical research standards, and community empowerment are essential components to enhance the prevention, detection, and control of zoonotic diseases, highlighting the interconnectedness of human, animal, and environmental health.

## 3. The Ethical Dilemmas of One Health: The Case of Animal Slaughter, Outline of One-Zoonoethics

In what follows, I briefly discuss the dilemma of mink culling in the pandemic scenario from the perspective of One Health. In 2020, the Prime Minister of Denmark, Mette Frederiksen, made the decision to sacrifice approximately 15 million minks due to a COVID-19 mutation identified on farms in the Nordic country. Denmark is the world’s leading producer of mink fur. The main argument in support of this decision was that the mutation detected among minks could infect humans and put the effectiveness of a new vaccine at risk. As illustrated in this case, certain anthropocentric measures to control zoonoses could give rise to ethical dilemmas related to the improper treatment of animals when human health is at risk. In order to resolve ethical dilemmas between competing value claims, we need to deepen the debate on multispecies justice [229,230,231,232,233,234,235,236,237].

To prevent animal slaughter and killing from becoming the new normal in pandemic situations, governments and institutions must continue to strive to incorporate multidimensional ethical analysis to understand how socially and culturally embedded values influence decision-making at different policy levels [238]. Furthermore, Biocultural education is pivotal in the One Health approach as it integrates traditional and scientific knowledge, fostering a holistic understanding of human, animal, and environmental health interactions. It promotes intercultural and multidimensional collaboration, enhancing disease prevention and response strategies through a more inclusive and effective management of global health risks [46,239]. It is essential to precisely articulate how the values and interests of animals, ecosystems, and ecological communities are incorporated into the One Health framework. Additionally, we must address critical questions, such as: What constitutes health and well-being for forests, ecosystems, or landscapes in relation to both human and non-human communities? Furthermore, what indicators can be used to assess whether a natural or constructed ecosystem maintains its health or has deteriorated? 

Animal ethics and environmental philosophy can significantly contribute to One Health by providing a framework for understanding and addressing the moral implications of human-animal-environment interactions. They promote a holistic approach that values non-human life and ecosystems, fostering ethical considerations that guide more inclusive and sustainable health practices [240,241,242,243]. 

To advance the revitalization and acceleration of One Health, we need to develop a long-term vision of the One Health policy that can positively value “entangled empathy” and relational vulnerability between different species [100,101,102,244]. In this part, I argue that, in order to responsibly address ethical and political issues related to the slaughter of animals in epidemic or pandemic emergencies, reactive measures based on crude utilitarianism are not enough; rather, we need to deeply rethink the practices and imaginaries associated with the mass slaughter of animals and develop new political processes, programs, and management tools to ensure that justice can be extended to both human and nonhuman animals [245,246].

In the case of zoonotic diseases, the moral meaning that people attribute to animals is context dependent; the statements and emotions for and against animal culling change meaningfully depending on the uncertainty and risks involved: the decisions made in a “normal scenario” are not the ones the same as those made under catastrophe scenarios. As Van Herten et al. [247] (p. 30) state: “In cases where human health is at risk, most people justify the culling of healthy animals. In situations where there is no danger that humans become infected, culling is less accepted”. The effectiveness and impact of measures to address emerging and re-emerging zoonotic disease risks depend on the belief systems, values, and assumptions of individuals and communities. Many times, unproven beliefs concerning wildlife and anthropocentric value schemes can become strong agencies that hinder the adoption of holistic health measures from a comprehensive One Health perspective [53,149,248,249].

Since 2008, The American Veterinary Medical Association established the concept of One Health, envisioning the possibility of fighting for the maintenance of certain levels of “optimal health” and establishing win-win solutions for all parties involved: human life, nonhuman life, and environments: “One Health is the collaborative effort of multiple disciplines—working locally, nationally, and globally—to attain optimal health for people, animals and our environment” [250]. This definition has evolved over time, particularly to address critiques of a superficial and anthropocentric view of One Health [18]. However, controversies persist, especially from ethicists who advocate for a deeper and broader understanding of animal welfare and rights. The One Health approach recognizes the interdependence and complex interconnectedness among humans, animals, environments, and the entire planet’s biosphere. Nevertheless, as Johnson et al. [251] (p. 185) notes, “although OH approaches commonly mention animal welfare, they have given minimal attention to animal health for the sake of animals. The anthropocentrism for which OH has been criticized instrumentalizes animals by recognizing their value only insofar as it contributes to human flourishing, thereby reinforcing the very anthropocentrism that justifies exploitation of farmed animals, encroachment on animal habitats, and the wildlife trade that OH purports to address”.

One Health scholars must take these legitimate criticisms very seriously and strive to address them. Only the implementation of a comprehensive and non-anthropocentric One Health vision can accomplish the promise of discovering fair and equitable solutions considering the vital interests of all parties involved, including both humans and nonhuman entities. As highlighted by Meisner et al. [252]: “Biomedical reductionism in One Health has resulted in a focus on human health threats from animals”. This demands a clarification of the concepts of interdependence, complexity, and interconnectedness that are at the core of One Health [158,159,253]. To build networks of care, solidarity, and kinship among nonhuman animals, communities, and environments, it is essential to reevaluate the normative and descriptive content of health and to consider ecological goods precisely as “common goods” rather than “natural resources”, a term still used by some One Health scholars [254]. This involves recognizing and accepting as legitimate the value claims of various actors, both human and nonhuman, who are connected to an intergenerational network of interests [237].

As Nussbaum [236] and Pelluchon [255,256] recently emphasized, the unjustified suffering and harm inflicted on animals, such as in the mass abandonment of pets or large-scale animal slaughter, although often normalized, reflect the extreme violence, arrogance, and indifference toward both human and nonhuman life. And all these forms of violence “are also injustices: we grant ourselves absolute sovereignty over sentient beings whose ethological needs and subjectivity should limit our right to exploit them as we see fit” [255] (p. 21). In the case of mink, the decision in favor of animal culling could be justified under the “harm principle”, initially introduced by Mill [257], which authorizes, under exceptional circumstances in which certain human assets are at serious risk, the ‘legitimate’ use of violence. The crucial point here is to precisely determine what is truly at stake when a government attempts to take exceptional measures to protect economies and public health [258,259,260,261,262]. This decision, based on political exceptionalism, created a domino effect and spread to other countries such as Holland, Sweden, Greece, and Spain [263,264]. This case illustrates the need for zoonoethics articulated to a comprehensive, multidimensional, and non-anthropocentric vision of One Health [29,30,147,228]. In the case of the animal culling during human pandemics, we must face two relevant ethical questions: First, is this an attitude of caution or an overreaction driven by panic? Second: Who loses and who wins after the sacrifice of animals that are conceived as an “extra factor” in the global fur industry, with an annual value of more than 22 billion dollars a year [265,266,267]?

To address the issue, it is necessary to reactivate the ethics of care and the profound and genuine well-being of wild, farm, and companion animals. Living in a respectful and just manner with nonhuman animals is possible and represents the most cost-effective long-term solution. However, we need to consider a zoonoethics of sustainability and care by ensuring the well-being of farm and wild animals, which are reservoirs of zoonotic agents, and by immunizing them against pathogens whenever possible. Zoonoethics reminds us that, to address the risks of infectious diseases, we need a healthy dose of prudence, caution, and common sense. As emphasized by Halabowsk and Rzymski [268] (p. 4): “The COVID-19 pandemic is the evidence that mink farming for fur represents a potential target of coronavirus spillovers and, consequently, a source of infection in humans who have contact with the animals. The ultimate solution to this is the development of an effective vaccine against SARS-CoV-2 in mink”. Finally, it is necessary to articulate a cautious perspective of the precautionary principle to zoonoethics to prevent control methods for managing zoonoses from bringing unforeseen consequences and unwanted harm to humans, animals, and environments [135,269,270,271].

As warned by Resnik: “Some of the methods used to prevent mosquito-borne diseases, such as draining swamps and spraying pesticides (especially DDT), can have adverse environmental impacts, such as the destruction of habitats and species” [272] (p. 2). Other cases that can illustrate the ethical dilemmas within One Health are the sacrifice of healthy surplus farm animals [243] and euthanasia or the “good death” of animals at zoos [187,273]. These cases, as seen by several scholars, are complex and go beyond the traditional tension between the positions of “welfarists” and “animal rights advocates”.

Table 2 presents policy recommendations aimed at overcoming ethical dilemmas associated with animal culling during pandemics. The first recommendation emphasizes the need to develop alternatives to culling, such as vaccination or quarantine measures, to reduce the need for mass culling and minimize harm to animal populations. Strategies to prevent the immoral slaughter of minks and other animals, such as phasing out the fur industry or implementing vaccination programs, are not always well received by all audiences. Sometimes, the economic argument is prioritized: “maintaining and investing in the well-being of farm animals can be very costly” [274], but failing to act or acting reactively can lead to increased costs and heightened risks at the complex human–nonhuman–environment interface [275]. The second recommendation focuses on strengthening ethical review processes to ensure that decisions involving animal culling are evaluated through a comprehensive lens of multispecies justice and ecological impacts. This recommendation is backed by several scholars who stress the need for ethical and critical contextualized scrutiny in One Health interventions and policies [172,173].

**Table 2 vetsci-11-00394-t002:** Ethical guidelines for addressing ERIDs and avoiding unnecessary animal culling.

Guideline	Description	Problem/Challenge	Authors
1. Develop Alternatives to Culling	Invest in research and development of alternatives to animal culling, such as vaccination or quarantine measures.	Reducing the need for mass culling and minimizing harm to animal populations.	[261,276]
2. Strengthen Ethical Review Processes	Implement stringent ethical review processes for animal culling decisions, ensuring consideration of multispecies justice and ecological impacts.	Ensuring ethical considerations are thoroughly evaluated before making culling decisions.	[137,247]
3. Surveillance and Monitoring	Establish systems for early detection and tracking of zoonotic diseases to respond quickly and effectively.	Timely detection and response to emerging zoonotic diseases.	[18,111,163,277]
4. Research and Development	Promote interdisciplinary research and develop new vaccines and treatments to combat zoonotic diseases.	Addressing gaps in knowledge and developing effective interventions.	[278,279]
5. One Health Education and Ecological Awareness	Incorporate OH core competencies and strengthening educational programs, including zoonoethics and AMR ethics for children, youth, and future professionals across the curriculum.	Increasing public understanding and engagement in zoonosis prevention.	[280,281,282,283,284]
6. Policies and Regulations	Create and enforce regulations to manage human–animal interactions and support sustainable practices.	Implementing bioethical and legal frameworks to prevent zoonotic disease transmission.	[19,20,285,286]
7. Ethical Disaster Management and Global Cooperation	Develop and implement ethical guidelines and rapid response plans for health crises and natural disasters, protecting animal welfare and preventing zoonotic outbreaks.	Ensure preparedness, minimize harm, and respond ethically to zoonotic outbreaks during crises.	[150,287,288,289,290]
8. Open Science for Future Pandemic Resilience	Develop future-oriented OH policies that enhance data analysis capabilities to understand disease dynamics and ensure the availability, quality, and management of accurate data for evidence-based decision-making.	Open science can greatly enhance OH pandemic responses by enabling rapid data sharing and collaboration.	[291,292]
9. Engage Local and Indigenous Communities	Engage local communities in zoonosis prevention and control, while fostering intercultural dialogue by integrating indigenous perspectives into One Health.	Ensuring local communities have a voice and active role in zoonosis prevention efforts in all levels.	[293,294,295]
10. Sustainable and Resilient Health Systems (SRHS)	Health systems need to develop new capacities and build synergies with other sectors and organizations to address risks of ERIDs.	Build better, more climate-resilient and environmentally sustainable health systems.	[296,297,298]
11. Enhance Biosecurity Measures	Implement comprehensive biosecurity protocols in intensive farming operations, including regular health monitoring and rapid response plans.	Preventing the spread of infectious diseases within and between animal populations to avoid large-scale outbreaks.	[55,164,298]
12. Promote Sustainable and One Welfare Farming Practices	Encourage sustainable and farming practices guided by interspecies ethics, reducing animal density and improving living conditions to lower disease risk.	Mitigating the conditions that facilitate the spread of zoonotic diseases in high-density farming and livestock.	[299,300]

Table 2 includes additional policy recommendations aimed at preventing the repetition of mass culling in intensive animal farming. The final recommendation advocates for enhancing biosecurity measures, which involve implementing rigorous health monitoring and rapid response plans in farming operations. By preventing the spread of infectious diseases within and between animal populations, this approach aims to avoid large-scale outbreaks, as highlighted by several studies [20,21,165,209]. The end recommendation focuses on promoting sustainable farming practices, such as reducing animal density and improving living conditions. This strategy aims to mitigate the conditions that facilitate the spread of zoonotic diseases in high-density farming environments, as discussed by [301,302,303].

## 4. Antimicrobial Resistance at the Human–Animal–Environment Interfaces: A Call for Global Action to Face the Antimicrobial Resistance (AMR)

This section is a silent but urgent call for global action and inter-epistemic dialogue on antimicrobial resistance (AMR). First, the best way forward in implementing a OH approach is to act preventively under three core principles: precaution, eco-wisdom, and responsibility from a comprehensive One Health for all, including value claims and the interests of humans, animals, and environments [135,271]. As noted early by Potter [195,304], human and animal healthcare practices generate a large amount of waste and pollution that can harm the environment and have long-term and systemic effects on planetary health. The environmental impact of hospitals—due to plastics, syringes, waste, and high energy consumption—is increasingly problematic for managing zoonotic and ERIDs. This situation is further exacerbated by the challenge of antimicrobial resistance [305,306], as the high demand for medical supplies and energy-intensive practices contributes to pollution and creates conditions that can fuel the spread of resistant pathogens. Poultry production, climate change, and intensive livestock farming are clearly linked to the spread of zoonotic diseases [307,308,309]. Environmental crimes and unsustainable animal husbandry practices not only create conditions conducive to the transmission of pathogens between animals and humans, but also accelerate the emergence of antimicrobial resistance [310,311]. The intensive use of antibiotics in these production systems contributes to the evolution of resistant strains, further complicating the control of zoonotic diseases and posing a significant threat to global public health.

Second, today, we have solid evidence about the incredible complexity of the mechanisms of spread of ERIDs and zoonoses that originate largely from the same drivers of global climate change. The nexus between wildlife trafficking, biodiversity loss, and environmental crimes lies at the root of the increasing zoonotic outbreaks driven by land use changes, wildlife exploitation, and hunting [8,11]. Therefore, it is essential to incorporate the systematic study of environmental crimes within One Health environmentalism and to design multisectoral strategies to address ecocide [29]. From the One Health approach, it has become commonplace to say that addressing zoonoses and ERIDs requires greater communication, collaboration, and interdisciplinary work between zoologists, virologists, veterinarians, foresters, and environmental humanities scholars. However, in practice, collaboration and inter-epistemic dialogue between experts in veterinary science, epidemiology, wildlife, and biological sciences on one side, and bioethicists, environmentalists, policymakers, and interested communities on the other, remain quite low [312]. Therefore, we must continue strengthening efforts in One Health education and interdisciplinary training for various professionals and practitioners [254,284].

Another problem is that very few public policy experts understand the high complexity of the issues involved in zoonoses and the challenges of One Health from a complex and multispecies justice approach. As noted by Waltner-Toews [253] (p. 7): “However, few animal science researchers, standing in their hazard suits and masks in a conventional chicken broiler barn, have made the mental connections between poultry rearing and ducks flying overhead, or between agribusiness and viruses and bacteria for which the ducks are a quiet, accessible vehicle for long-distance air travel”.

At a meeting in March 2023, the Quadripartite made an emphatic “call to action for One Health for a safer world”. It calls for enhanced collaboration among sectors like human, animal, plant, and environmental health to create a safer, more resilient world. The call also highlights the need for integrated policy frameworks and investments to address health challenges at the human–animal–environment interface. The latest World Health Assembly held in Geneva this year concluded without a clear agreement on the conditions and scope of the Pandemic Accord [313]. This brings me to my third and final call. Perhaps one of the problems that best illustrates the importance of zoonoethics is the problem of AMR. Until approximately fifty years ago, the problem of AMR was basically understood as a problem of interest to microbiologists, virologists, and a small group of economists interested in the development of antibiotics and new drugs. Over the past two decades, significant efforts and transdisciplinary collaboration have highlighted that the indiscriminate use of antimicrobials is a major driver of antimicrobial resistance (AMR). This issue has emerged as one of the top ten ethical and political challenges facing humanity. AMR is closely linked to broader global crises, including food insecurity, poverty, lack of clean water and sanitation, inadequate primary healthcare, and failures in waste management and public health policies aimed at infection control [307,311,314].

The WHO has corroborated that all of these factors promote the spread of microbes, some of which may be resistant to antimicrobial treatment. This explains why the concept of antimicrobial stewardship has gained so much worth in recent years [315,316,317]. Reducing the AMR problem to a hospital issue would be irrational and could perpetuate injustices. The widespread use of bactericides in the care of humans, animals, and agriculture means that we are in the middle of a malignant circle of bioaccumulation of bacteria and microorganisms resistant to antibiotics. As pointed out by several authors, antibiotic-resistant bacteria (ARB) and antibiotic resistance genes (ARGs) are found beyond hospital settings, often originating from natural environments like soil and water. Environmental strains can transfer ARGs to human pathogens [317]. Therefore, identifying the sources, distribution, and anthropogenic factors is crucial for developing strategies to combat antibiotic resistance [71,73,80,171]. Table 3 presents a set of policy guidelines for addressing AMR comprehensively and with interspecies equity.

Finally, is important make a call to the Quadripartite and WHO to reinforce educational campaigns to inform the public and healthcare providers about the responsible use of antibiotics [337]. Increasing awareness about the consequences of antibiotic misuse and promoting proper prescription practices can reduce unnecessary antibiotic consumption and slow the spread of resistance [69,74,75,76]. An ethical approach to AMR must address the gaps and injustices that persist in equitable access to medications and medical treatments: [338] “Many people around the world still do not have access to antimicrobials. Ensuring equitable and affordable access to quality antimicrobial agents and their responsible and sustainable use is an essential component of the global response to antimicrobial resistance”. These recommendations provide a comprehensive framework to address AMR through an ethical and policy lens, incorporating the One Health approach and emphasizing the interconnectedness of human, animal, and environmental health.

## 5. Zoonoethics, Intercultural Dialogue, and Entangled Empathy: A Silent Call for Interspecies Solidarity

A careful, transdisciplinary study of the intricate interdependence among all forms of life can lead to better management of zoonotic diseases and guide us in acting prudently to prevent the next pandemic. In closing, I reiterate my earnest call to urgently undertake prudent and systematic actions to halt the destruction of rainforests and counteract the effects of climate change, which are causing significant damage to Arctic and Antarctic ecosystems. We all share this planet, and all forms of life, from the smallest organisms to the largest mammals, face the threat of extinction. The One Health approach can be grounded in the profound vision of interdependence and interconnectedness embedded in the intercultural wisdom of indigenous peoples and the ethnomedical practices of diverse local communities [31,34]. As the Biological Biodiversity Convention has recently emphasized: [34] (p. 6) “The relationship between biodiversity loss, the emergence and spread of communicable and non-communicable diseases and increasing health inequalities is well known, as is the role of conservation, restoration and sustainable use of biodiversity in prevention, reduction and proactive management of communicable and non-communicable disease risks”. However, there are still reasons for hope and joy. For instance, initiatives from southern Chile are advancing intercultural and interspecies encounters that broaden our traditional view of human agency. The Biocultural Education Program of the Omora Ethnobotanical Park in the Cape Horn Biosphere Reserve in Chile exemplifies life consortia between and beyond species and demonstrates ways to inhabit and cohabit environments responsibly and attentively [339].

By incorporating environmental philosophy [340,341], intercultural bioethics, One Health social sciences, and zoonoethics, we can broaden our concerns for planetary health from veterinary, epidemiology, and earth sciences to one of interconnectedness, entanglement, and complexity. The issue of epistemic injustice in bioethics is crucial for effectively addressing the risks of epidemics and pandemics. The question of who speaks, how they speak, and from where they speak—in other words, the question of testimony and the value of the voices, narratives, and struggles of Indigenous peoples, women, and local communities—is essential for a comprehensive response to zoonotic diseases. One Health approaches and policies cannot be imposed on local communities through draconian measures but must arise from consensus and constructive conflict. One Health researchers still have much to learn from the intercultural health practices of many Indigenous peoples in the Global South. Perhaps the most critical and challenging issue for One Health is addressing multispecies injustices, violence against women and girls, and ecological racism. Therefore, one of my final calls is to decolonize and gender the theory and practice of One Health. This is the only viable path to overcoming these injustices. My quiet plea is to advance along the path of interculturality and to envision ways of reparative and restorative justice for the life consortia between humans and non-humans [342,343].

As Waltner-Toews [253] states: “It is a grievous mistake to imagine that pandemics can be understood and managed by studying the pieces separately (viruses, birds, pigs, people). To understand the challenges of learning to live with diverse microbial populations, we need to re-imagine the world in deeper, more complex, more evidence-based ecosystem terms. It is one thing to document in detail the cellular and biochemical structure of dead ducks in a marsh in Saskatchewan, as well as, more recently, the microbiomes they carry, or those of dead chickens in a barn in British Columbia. It is quite another thing to understand the relationships among multiple species at multiple scales”. To deal ethically and prudently with emerging infectious diseases, we need to improve the intercultural collaboration between veterinary medicine, cutting-edge Western knowledge, and the ancestral knowledge of traditional systems of indigenous peoples [57]. All of these efforts would lead us to a more modest vision of ourselves and would help us move toward policies of solidarity, care, and interspecies cooperation [100,101,230]. Implementing the One Health approach necessitates integrating various diverse biocultural and symbolic universes and values, while also instigating constructive conflict and establishing the groundwork for intercultural and interreligious dialogue. This will allow us to craft fresh “interwoven narratives” that foster sustainable ways of inhabiting the future [231,233,234,344,345].

In order to tell and pass on stories of resilience and planetary health to future generations, it then becomes imperative to recognize the ignored interests and perspectives of value, the differences in political and economic power, and how gender impacts policies to confront AMR, climate change, and zoonoses. All this has a lot to do with multispecies justice. As stated by [253] (p. 7), “the real costs of producing low-cost chicken are being paid in economic subsidies to fossil fuels and corn, in lost biodiversity in Brazil and in the oceans, and in urgent adaptations to dramatic, unstable climate change”. As I previously showed in the case of mink, the cost of the utilitarian management model to prevent the spread of a potentially pandemic virus was borne by defenseless animals. Clearly, this is unfair, painful, and immoral. The costs of developing new products for the cosmetic industry and drugs to stop aging are being paid by animals that are used in laboratories as “test objects” that at some point become “disposable”. The costs of so-called development and postponing solutions to climate change are being paid by millions of animals that are slaughtered annually on farms to feed a growing population and by impoverished farmers and food industry workers. The costs of our immense loneliness and dementia are being paid for by the suffering we cause to animal species and populations that we are pushing to the brink of extinction [346,347,348,349]. Terrestrial and aquatic ecosystems, and large rivers, seas, and lakes, are also paying a high price for our immoral lifestyles and our excessive consumption of raw materials. It is absurd and shameful to see that the cost of wars, techno-scientific development, and testing of unconventional weapons in the oceans can be paid at a very high price by our children, grandchildren, and future generations of human and nonhuman animals. Thus, each generation can take part, almost unsuspected, in what Gardiner calls a “severe intergenerational tyranny” [349]. In order not to end with such a discouraging panorama,

From the heart of southern Chile, near the Cape Horn Biosphere Reserve, I make a quiet appeal: Environmentalists and One Health advocates, let’s unite to forge new synergies, enhance cross-cultural dialogue, and drive the changes needed to halt the systematic destruction of wildlife. We urgently need a new form of global ecological cooperation and solidarity to tackle the immense challenges of the Sixth Mass Extinction and address health issues, inequity, structural poverty, and conflicts that are closing the window of opportunity to reverse the course of extinction. But it is still within our reach to make a difference.

If “everything is related to everything else”, as the American biologist and environmentalist Barry Commoner also thought [155], then we still have the possibility of building what Lori Gruen [101] calls “the entangled empathy”, which is not only the impulse to go out to meet the other or try to “put oneself in the shoes of the other who suffers” but also requires a motivational effort. Changing our expectations and views of animals requires a conscious and intentional effort on our part. (p. 80). Entangled empathy has an emotional, cognitive, and psychological component, perhaps in the Aristotelian sense of cultivating a set of virtues and emotional responses such as solidarity, respect, and understanding, especially when we lack relevant information about the intentions, values, and belief systems at the table. Thus, entangled empathy has to do with our dispositions or habits; it also implies “moral courage” and practical-motivational components. Gruen [101] adds: “Entangled empathy requires a certain amount of perspective taking. In this sense, it is more akin to some of the other ways of understanding empathy. Perspective taking means trying to get outside of your own sensibility even though of course we are limited, in some ways, by our own perspectives on the world” (p. 81). In the case of the prudent management of zoonoses, contextual observations are relevant because they help establish guidelines for action: For example, in the case of the COVID-19 pandemic, great panic and certain feelings of anger, fear, and hatred towards bats and pangolins, which were considered scapegoats at the beginning of the pandemic; many media outlets and some politicians also inoculated their audiences with feelings of retaliation and revenge toward “the Chinese” and even toward wet markets and wildlife as a whole. Gruen [101] (p. 81) stressed: “With some animals, dangerous wild animals, for example, it is not going to be possible or wise to get close to those individuals. However, there are usually trained ethologists and zoologists who study these individuals”. When dealing with certain wild animals, such as lions, whales, bears, bats, and others, empathetic treatment means staying as far away from their environment as possible and learning to appreciate the network of interspecies assemblages of their ecological communities from a healthy distance. To do this, it is not needed do a thorough analysis, but rather exercise our common sense and follow our “best judgment”. Experts in wildlife management could tell us storytelling about how they learned to live with them, and the narratives also become a powerful resource to enhance entangled empathy and prudent treatment toward wildlife.

Finally, Gruen [101] hits the bullseye when she says (p. 81): “The process of entangled empathy then involves emotion, imagination, cognition, justice and care. it focuses on flourishing there is also room in the process for emotions like outrage and indignance and anger and shame. When we recognize that other human beings and other animals are not flourishing, when we recognize that other animals and other human beings are thought to be disposable and killable”. In short, a zoonoethics that aims to provide guidelines for prudently managing zoonoses and EIDs must recover the idea that all forms of life are valuable in themselves and are finely intertwined. No form of human or nonhuman life on planet Earth should become something “sacrificeable”, something “disposable”, or something “superfluous”. I believe that this type of ethical stance and this mood are what should accompany and encourage future developments in zoonoethics.

## 6. A Call for Urgent Action: Policy Recommendations for an Inclusive, Intercultural, and Gender-Sensitive One Health Approach

In order to support a comprehensive and multidimensional approach to One Health, I present in this section a set of policy guidelines and ethical recommendations:Advance international legislation to recognize the international crime of ecocide and crimes against biodiversity, not only as circumscribed and peripheral damages that affect human health but as damages that affect the health of humans, animals, and environments and constitute an attack against future generations.Stop the illegal wildlife trade, the slaughter of wild animals in wet markets, and the illegal timber trade in tropical forests, especially in areas of high biodiversity in Asia, Africa, and the Americas. This requires deepening the partnership against wildlife crime and developing new intercultural capacities to reconnect human and nonhuman animals, places, and the planet.We need closer transdisciplinary collaboration, including gender and intercultural perspectives, to study the socioeconomic, cultural, and environmental determinants and drivers of zoonotic diseases: “Developing a multi-sectoral preparedness and response plans for control of zoonotic diseases through a comprehensive risk assessment, improving laboratory diagnostic capacities, joint surveillance activities at the animal-human interface” [111,350].Strengthen political commitment, national planning, and regional coordination mechanisms; this requires working towards a One Health approach based on principles of intersectionality, interculturality and global solidarity. These plans and long-term strategies should be evaluated from a complexity approach at the local, regional, and global levels [66,288,290].Promote equitable and long-term synergies between Western health systems and local and indigenous community health knowledge systems and practices. Additionally, we need to create innovative strategies and establish regional and global information networks to facilitate knowledge sharing and enhance collaborative efforts to manage risks across the various interfaces of One Health. In particular, the wildlife–livestock–human interface is one of the areas of greatest risk and vulnerability.Promoting a One Digital Health approach: Europe, the United States, and other high-income countries have strong epidemiological surveillance systems that provide access to comprehensive data, tables, and maps on infectious diseases, but low- and middle-income countries in regions such as South and Central Asia, Africa, and Central and South America do not yet have robust surveillance systems to develop systemic preparedness, mitigation, and prevention plans and strategies for zoonotic diseases.Given the growing importance of AI and accelerating digitalization in health systems, countries and regions should develop synergies and share resources to overcome gaps in access to resources through a One Digital Health equity approach [277,351,352].One of the challenges highlighted by the COVID-19 pandemic was the need to work together across sectors and regions to develop greater North–South synergies of cooperation, equity, and multispecies justice to lay the foundations for a sustainable One Health system based on a broad vision of health, common goods, and eco-solidarity.

## 7. Conclusions

In examining the historical development of infectious diseases, scholars have demonstrated that the impact of these illnesses extends well beyond the morbidity and mortality indicators associated with epidemics [126,353,354,355,356,357]. The history of emerging and re-emerging infectious diseases frequently coincides with the history of social inequities and the social construction of specific communities, groups, and regions as “vulnerable” and primary sources of infectious risk. As Rushton [330] states (p. 121): “The impacts of disease have included inducing political and social instability (in extreme cases even contributing to the collapse of entire civilisations, as occurred when Amerindian societies were devastated by smallpox), causing migration as people attempt to ‘get out of harm’s way’, undermining economies, and playing a part in determining the course of armed conflicts. Globalisation, however, is generally understood to have fundamentally changed the nature of the cross-border disease threat as a result of the more extensive, and more rapid, international movement of people, animals, food, and other goods”. It is evident that globalization, economic growth, extractivism, and the acceleration of large flows of humans, animals, and goods due to global transportation in the capitalist economy are significant contributors to the emergence and spread of infectious diseases. “But whilst globalisation-related changes have no doubt sped up the geographical movement of pathogens, such spread was always a reality of life on earth. As a result, fear of the importation of pathogens from a dangerous outside world has been a topic of political discussion and action for centuries” [330] (p. 121). Today, the fears and new biophobias raised by COVID-19 and the climate of economic and geopolitical instability have undermined the efforts of many organizations to move toward One Health Diplomacy [329] and advance the construction of a global Pandemic Agreement to prevent, control, and respond to emerging zoonotic disease risks [294].

In conclusion, it is imperative to address the pervasive phenomenon of fear, disinformation, and parsimony that has become an obstacle to achieving viable and sustainable futures for human and nonhuman communities. To this end, it is crucial to integrate One Health initiatives with studies on power and security dynamics. As Price-Smith [127] emphasizes (p. 189): “The role of infectious disease in modern security studies originates in the historical accounts of Thucydides, Machiavelli, and Rousseau, all regarded as republican progenitors of the political paradigm known as Classical Realism”. With relative ease and in a domino effect, public health policies during times of fear, economic decline, and disaster scenarios can be guided by political exceptionalism leading to draconian measures based on political realism rather than a One Health ethics [358,359,360]. As highlighted by the COVID-19 pandemic, policy decisions for managing infectious diseases in disaster scenarios are not neutral; they are based on visions, values, and imaginaries of health and disease that emerged in modernity. It is therefore imperative to persist in elucidating the ethical principles and policies that guide global political decision-making in a world characterized by complexity and uncertainty. Moreover, sustained collaborative efforts are essential to advancing holistic, inclusive, and effective ‘One Health’ initiatives. Reconnecting with the nonhuman world and the ecosystems we inhabit is crucial to fostering a forward-thinking zoonoethics, as proposed in this paper [361].

## Figures and Tables

**Figure 1 vetsci-11-00394-f001:**
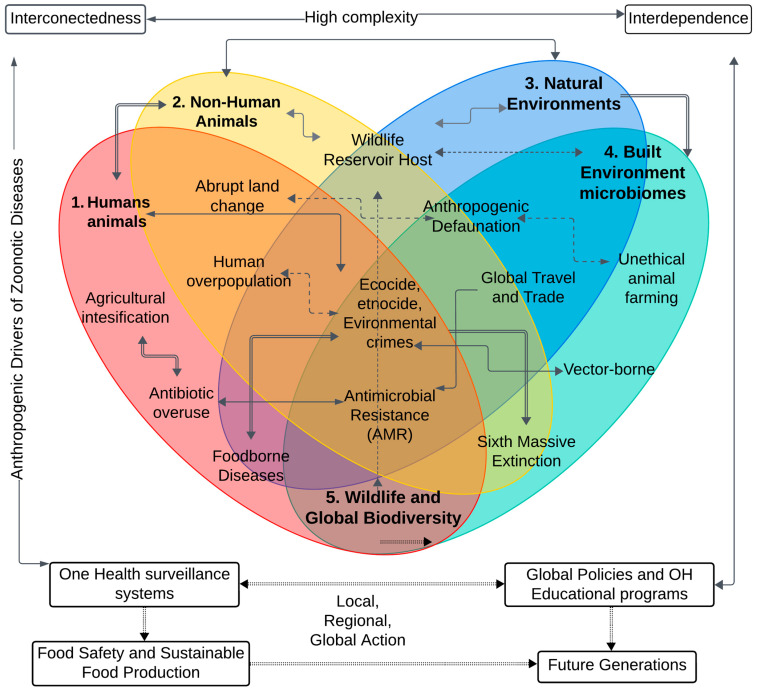
Illustrates the intricate interdependencies and linkages that underpin the health of our planet. This model emphasizes several critical aspects, including: (i) the pervasive use of antimicrobials across various stages of intensive plant and animal production system; (ii) the intensification of agriculture, which is closely tied to global population growth, abrupt land use changes; and (iii) the progressive loss of wildlife and anthropogenic deforestation, which are interconnected with political and social phenomena such as ethnocide, ecocide, and environmental crimes. To highlight the different interfaces involved in pathogen transmission, the color coding of the diagram was intentionally chosen: (1) Humans and Animals (Red): This section underscores the impacts of agricultural intensification and antibiotic overuse, which are closely tied to human activities and have profound consequences on health and food security. (2) Non-Human Animals (Yellow): Highlighting the role of non-human animals, particularly wildlife, this area focuses on issues such as abrupt land change and human overpopulation, which contribute to the displacement and stress on wildlife, increasing the risk of zoonotic disease transmission. (3) Natural Environments (Blue): This interface addresses the importance of natural environments, drawing attention to anthropogenic defaunation and the role of wildlife as reservoirs for zoonotic pathogens. It emphasizes the need for conservation to maintain ecological balance. (4) Built Environment Microbiomes (Teal): This section deals with the impacts of the built environment, including unethical animal farming and the spread of vector-borne diseases. It highlights how human-made environments contribute to the spread of diseases and affect global health. (5) Wildlife and Global Biodiversity (Purple): Central to the diagram, this section illustrates the global significance of biodiversity. Issues such as antimicrobial resistance (AMR), foodborne diseases, and the sixth mass extinction are shown to be critical factors that interlink with all other components. The dotted lines and arrows within the diagram illustrate the deep interconnections and interdependencies that permeate these components, reflecting the intricate web of factors influencing zoonotic diseases. The outer boundaries reinforce the notion that these elements are governed by dynamics of high complexity, necessitating a holistic and integrated approach to addressing planetary health challenges.

**Figure 2 vetsci-11-00394-f002:**
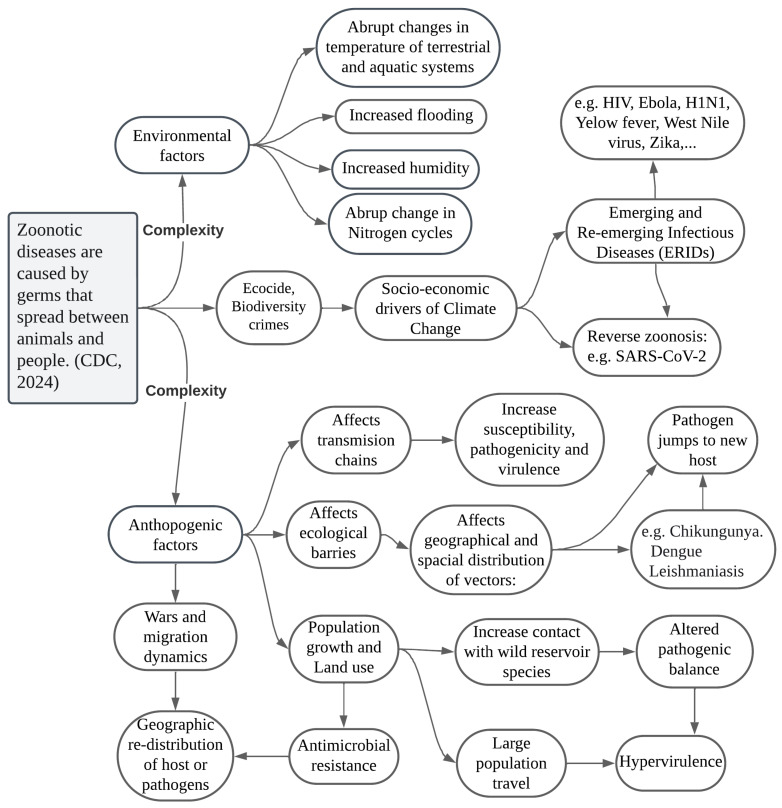
The interrelationship of events that trigger the onset of infectious diseases.

**Table 1 vetsci-11-00394-t001:** Bioethical principles and political guidelines for addressing emerging and re-emerging infectious diseases.

Ethical Principles	Zoonotic Ethics Guidelines	Problem/Challenge
Autonomy	Respect the autonomy of communities and individuals in One Health decision-making.	Ensuring individual and community rights are upheld during public health interventions.
Beneficence	Implement measures that maximize well-being and minimize harm for both humans, animals, and environments.	Balancing the benefits of interventions against potential risks and harms.
Non-maleficence	Avoid actions that cause unnecessary harm, such as the indiscriminate culling of animals without comprehensive ethical analysis.	Preventing unethical practices that could cause harm to animals and ecosystems.
Justice	Ensure the equitable distribution of resources and treatments to prevent and control zoonotic diseases. Ensuring universal access to health innovations and vaccines.	Addressing disparities in access to healthcare and resources for disease prevention and control.
Ethical Deliberation/ Constructive Conflict	Consider all relevant interests at stake and integrate the value and welfare of human animals when implementing management plans and measures to control zoonoses. Strategies: Multi-layered and multi-actor assessment. Context-dependent analysis	Mitigate the negative impacts and side-effects on nonhuman animals of strategies and plans to control zoonoses and epidemic risks.
Environmental Responsibility	It is imperative to adopt responsible practices in all sectors that safeguard ecosystems and biodiversity, acknowledging their integral role in One Health and the secure survival of future generations.	Mitigating the impact of human activities on ecosystems and preventing biodiversity loss.
Transparency and Communication	Maintain open and honest communication with the public about risks and measures taken to control zoonoses.	Building public trust and ensuring informed participation in public health measures.
Respect for Life	Value and protect the lives of all living beings, recognizing the interdependence between humans, animals, and environments.	Promoting a holistic view of life that includes the well-being of all species.
Solidarity and Cooperation	Promote international collaboration and solidarity among nations to combat global zoonotic threats.	Fostering global cooperation to address transboundary zoonotic disease threats.
Precaution and Prudence	Adopt preventive and prudent measures in the face of scientific uncertainty and potential risks of new zoonoses.	Taking proactive steps to prevent outbreaks even when full scientific certainty is not available.
Intergenerational Equity	Make decisions that do not compromise the health and well-being of future generations. Avoid practices and policies that transfer risks and damages to future generations. Inter/trans-generational justices.	Ensuring sustainable practices that do not deplete resources or harm future generations.
Care	It is imperative that robust legislation be enacted to address the crime of ecocide, including the introduction of criminal laws to deter and punish the systematic destruction of biodiversity and the trafficking of wildlife.	Mitigation of large-scale species extinction and crimes against biodiversity

**Table 3 vetsci-11-00394-t003:** Ethical and policy recommendations for addressing antimicrobial resistance (AMR).

Category	Recommendation	Rationale
Preventive Action	Accelerate the implementation of the One Health approach from a preventive and anticipatory lens. Shifting from reactive to proactive zoonotic risk mitigation model is essential.	Implement a preventive One Health strategy with a precautionary approach helps to address the root causes of AMR and promotes sustainable practices [135,318].
Biosecurity Processes	Ensure that all medical and veterinary waste undergoes effective decontamination processes, such as autoclaving or chemical disinfection, to eliminate pathogens and reduce the risk of AMR.	Proper waste management and biosecurity tools minimizes the ecological footprint of animal and healthcare practices, reducing environmental pollution and the spread of AMR [319].
Sustainable Food and Agricultural Practices	Promote sustainable agriculture and livestock practices, reducing antimicrobial use in food production to prevent resistant pathogens.	Sustainable practices prevent AMR by reducing unnecessary antimicrobial use and promoting ethical treatment of animals and the environment [317,320,321,322].
Public Policy	Educate public policy experts on the complexity of zoonoses, EIDs, and AMR from a One Health and multispecies justice perspective.	Informed policymakers can develop more effective policies to address AMR, considering its broader impact on public health and the environment [253,323,324].
Antimicrobial Stewardship	Promote interdisciplinary and Antimicrobial Stewardship programs to ensure the responsible use of antimicrobials in human and animal healthcare, agriculture, and food-chains.	Antimicrobial Stewardship programs help mitigate the overuse and misuse of antimicrobials, reducing the development of resistant pathogens [319,325,326,327].
Infrastructure Improvement	Improve infrastructure for clean water, sanitation, and waste management to prevent the spread of resistant microbes.	Improving infrastructure tackles AMR’s root causes in low-resource settings by addressing inadequate sanitation and waste management [322,327].
Ethical Reflection	Encourage ethical reflection on healthcare practices, animal treatment, and environmental impact in decision-making processes.	Ethical considerations ensure that actions taken to address AMR are just, sustainable, and responsible, benefiting all stakeholders involved [67,68,70,72].
One Health Education and Antibiotic Awareness	Increase public awareness and education about the causes and consequences of AMR and the importance of responsible antimicrobial use.	One Health education promotes responsible behavior and support for AMR-reducing policies, making ethical reflection on AMR crucial for a sustainable and just future [77,328].
Global Cooperation	Strengthen international cooperation and coordination in monitoring, research, and response to AMR.	Global cooperation is crucial for addressing AMR, as it is a transboundary issue that requires coordinated efforts across countries and regions. [329,330,331].
Food Chains and Water Systems	Strengthen and enforce regulations on the use of antibiotics in agriculture and aquaculture.	By regulating the use of antibiotics in food production, we can reduce the development of antibiotic-resistant bacteria in the environment, which can be transferred to humans through the food chain [323,324,332,333].
Multispecies Justice	Implement comprehensive OH policies that prioritize the welfare and the intrinsic value of all including environments, human, and nonhuman animals affected by AMR.	Addressing AMR ethically requires considering the impact on diverse species and ecosystems, ensuring fair treatment and health outcomes for all [253,334,335].
One Welfare	One Welfare by integrating animal, environmental, and public health in antimicrobial stewardship policies.	Integrating this approach into AMR strategies enables sustainable, equitable solutions across all sectors [299,300,336].

## Data Availability

Not applicable.

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
