# Peer review of "One Health Ethics and the Ethics of Zoonoses: A Silent Call for Global Action"

_vetsci, 2024, doi:10.3390/vetsci11090394_

Round 1

Reviewer 1 Report

Comments and Suggestions for Authors

The most interesting part of your review is the introduction of the term ‘zoonoethics,’ and my main comment is regarding this part of the paper (pages 7-10). I would recommend clarifying the relationships between the proposed term ‘zoonotics’ and multiple related terms, which you used in this section. These terms include ‘bioethics,’ ‘one bioethics,’ ‘ecological bioethics,' and ‘One Global Bioethics.’ This would help readers appreciate the proposed term. You skip another term – ‘zooethics,’ whereas the question of “euthanasia or the good death of animals at zoos" is more related to this discipline.

I have only a few specific and minor comments:

Line 184: The statement that “companion mammals are genetically more similar to human” sounds very strange.

Lines 193-194: The new infections caused by previously unknown pathogens are called ‘emergent,’ not ‘re-emergent’ ones.

Line 204: Geographic expansion of ticks also depends on temperature and humidity.

Lines 510-511: This sentence looks more like a personal comment than a part of the paper. It’s better to delete it.

Line 707: Correct this sentence. You have the word ‘punitive’ after the dot.

Author Response

Comment 1: I am appreciative of these nuanced observations.. I concur with the recommendation to elucidate the epistemic scope of the term "zoonoethics," which I am introducing for the first time in the academic literature. I will integrate an additional paragraph into the paper that delineates the interrelationship between this term and "zooethics," as well as their shared concerns with the broader field of global bioethics and, more specifically, with One Biethics. 
Comment 2: The minor inaccuracies in the highlighted lines have already been corrected in the final version. 

Reviewer 2 Report

Comments and Suggestions for Authors

This is a broad and ambitious analysis of zoonosis based on 308 actual publications.

In this review the author makes a throughout and critical analysis about the concept of zoonoses in the present world, from an ethical and political standpoint. Besides the simplistic human and veterinary medicine interactions, other factors like environment, social culture, economy, policy, anthropogenic climate change, and antimicrobial resistance are analysed, and show Zoonoses as a very complex issue. The author makes also suggestions, recommendations and personal opinions to address different aspects of society in the frame of zoonoses.

The review is well organized, although the complexity of the approach detailing several aspects, made it a dense document. 

Author Response

Comments 1: I appreciate the reviewer's comments. In order to avoid making the bibliography too long, I will remove from the paper those that are not strictly necessary to illustrate the central arguments.

Comments 2: I will try to remove arguments that I consider dense without compromising the complexity approach that underlies the central position of the paper. 

Reviewer 3 Report

Comments and Suggestions for Authors

Dear Author,

the critical review you submitted to the Veterinary Sciences journal aroused in me great interest and boundless admiration for all the work done, especially considering that you did it alone, from the design of the study to the drafting of the manuscript.

I believe that to carry out this work it is necessary to embody the concept of One Health and possess inter- and multidisciplinary training and culture, the latter aspect that all scientists should have to define themselves as such.

Just some of my short suggestions to make your manuscript definitively publishable:

- I would add some keywords (e.g. AMR, policy recommendations, etc.), considering that your critical review is very rich in terms that are not easily found in scientific articles on the topic of zoonoses; furthermore, the issues addressed are numerous and complex;

- the references must be inserted in the text with numbers, while in the relevant "References" section the list must follow the order of appearance in the main text (see Vet Sci template);

Finally, I believe that your critical review represents numerous elements of reflection for all those who, like me, profess the complexity of the One Health approach, but also an interesting topic of reading and study for all those who are training to become future (one) health professionals.

At the moment, in my opinion, the manuscript is "accepted after minor revision".

Kind Regards

Author Response

Comments 1: I appreciate the positive and constructive comments of the reviewer. I agree that future One Health professionals should be well equipped with ethical competencies to face interdisciplinary challenges in a changing and dynamic world. 

Comments 2: I appreciate the words you suggest and will include them in the keywords to help readers and audiences find the paper and continue to search for related work. 

Comments 3: I will be sure to adjust the references according to the rules of Vet Sci.